# ToBRFV Infects the Reproductive Tissues of Tomato Plants but Is Not Transmitted to the Progenies by Pollination

**DOI:** 10.3390/cells11182864

**Published:** 2022-09-14

**Authors:** Ben Avni, Dana Gelbart, Tali Sufrin-Ringwald, Hanita Zemach, Eduard Belausov, Rina Kamenetsky-Goldstein, Moshe Lapidot

**Affiliations:** 1Institute of Plant Sciences, Volcani Center, Agricultural Research Organization, Rishon LeZion 7505101, Israel; 2The Robert, H. Smith Faculty of Agriculture, Food and Environment, The Hebrew University of Jerusalem, Rehovot 7610001, Israel

**Keywords:** ToBRFV, *Tobamovirus*, FISH, tomato, flower, seed, pollen

## Abstract

Tomato brown rugose fruit virus (ToBRFV), a newly identified *Tobamovirus,* has recently emerged as a significant pathogen of tomato plants (*Solanum lycopersicum*). The virus can evade or overcome the known tobamovirus resistance in tomatoes, i.e., *Tm-1*, *Tm-2*, and its allele *Tm-2*^2^. ToBRFV was identified for the first time only a few years ago, and its interactions with the tomato host are still not clear. We investigated ToBRFV’s presence in the reproductive tissues of tomato using fluorescent in situ hybridization (FISH) and RT-PCR. In infected plants, the virus was detected in the leaves, petals, ovary, stamen, style, stigma, and pollen grains but not inside the ovules. Fruits and seeds harvested from infected plants were contaminated with the virus. To test whether the virus is pollen transmitted, clean mother plants were hand pollinated with pollen from ToBRFV-infected plants and grown to fruit. None of the fruits and seeds harvested from the pollinated clean mother plants contained ToBRFV. Pollen germination assays revealed the germination arrest of ToBRFV-infected pollen. We concluded that ToBRFV might infect reproductive organs and pollen grains of tomato but that it is not pollen transmitted.

## 1. Introduction

Tomato (*Solanum lycopersicum* L.) is one of the most important vegetables grown globally. A major constraint in tomato cultivation is infection by viruses, resulting in decreased fruit quality and yield reduction [1]. Tomato brown rugose fruit virus (ToBRFV), a newly identified *Tobamovirus,* has recently emerged as a significant pathogen of tomato plants. The virus was first identified in 2015 in Jordan when a new disease of tomatoes was detected [2]. The causal agent was transmitted mechanically and was recognized as a new tobamovirus. During the autumn of 2014, a new viral disease broke out in greenhouse-grown commercial tomato hybrids in Israel [3,4], all of which carry the tomato mosaic virus (ToMV) resistance gene *Tm-2*^2^. The virus was identified as a tobamovirus, and its nucleotide sequence was 99% identical to the Jordanian isolate of ToBRFV [3]. ToBRFV-infected tomato plants express severe disease symptoms, including a pronounced mosaic of leaves and leaf deformation and elongation up to becoming thin threads. Disease symptoms also appear on tomato fruits and include yellow and orange marbling and, at times, the appearance of dark “brown rugose” spots on the fruits, making the fruits unmarketable [2,3].

Following its identification a few years ago, the virus has quickly spread globally and is now present in a large number of tomato-growing countries. These include Spain, Italy, and Greece in Europe [5,6,7]; Egypt in Africa [8]; China and Turkey in Asia [9,10]; the USA and Canada in North America [11,12]; and Mexico in Central America [13].

The virus genome comprises a single positive-sense single-stranded RNA molecule, nearly 6400 nt in length. The virus codes for the four typical tobamovirus proteins: two subunits of the viral replicase, a movement protein, and a capsid protein. The highly stable tobamovirus virion has a rod shape and is viable in the environment for an extended period of time [3,14]. Like other tobamoviruses, ToBRFV has no insect vector and is transmitted mechanically by contact, mainly by infected plants, plant debris, contaminated soil and tools, and farm workers handling the plants.

It is well documented that seeds harvested from tobamovirus-infected plants such as ToMV-infected tomato or cucumber green mottle mosaic virus (CGMMV)-infected cucumber plants are contaminated with the virus [15,16]. Moreover, in these crops, tobamoviruses can be transmitted by contaminated seeds (either untreated or insufficiently treated with disinfectants) to the growing seedling [16,17]. Similarly, seeds harvested from ToBRFV-infected tomato plants are contaminated with the virus [18,19,20,21,22]. ToBRFV is localized on the seed coat, sometimes in the endosperm, but it was not found in the embryo [18,19,22]. Several disinfection treatments were tested for ToBRFV removal efficiency from the contaminated seeds. It was found that disinfection treatments (such as incubation in 2% HCL for 30 min or 10% TSP for 180 min) removed the contaminated ToBRFV virions from the seeds without affecting the germination rate of the treated seeds [18,20,21]. However, since the virus is transmitted by contact, it has been suggested that the virus can transmit from the contaminated seeds to the newly germinated seedlings. Grow-out experiments demonstrated low seed-to-seedling transmission rates, ranging from 0.08% [22] to 1.8% [18]. Consequently, it was concluded that ToBRFV is seed-borne (located externally on the seed coat) in tomatoes and is transmitted mechanically from contaminated tomato seeds to the seedlings.

Additionally, it was found that some tomato fruits sold in open markets or supermarkets were infected with ToBRFV. Moreover, damaged fruits could be an effective inoculum for virus transmission [19].

Tomato pollinators such as bumblebees (*Bumbus terrestris*) were also able to transmit ToBRFV during the buzz pollination of the flowers [23]. Clean beehives placed near infected tomato plants became contaminated with ToBRFV. When ToBRFV-contaminated beehives were placed in a greenhouse containing only clean tomato plants, the bumblebees infected the plants with ToBRFV [23].

ToBRFV infects tomato plants that harbor the known genetic resistances against tobamoviruses, namely *Tm-1*, *Tm-2*, and its commercially widely used allele *Tm-2*^2^ [3,24,25]. Recent work shows that the *Tm-2*^2^ resistance fails to recognize the movement protein of ToBRFV—thus, the virus evades the *Tm-2*^2^ resistance [26]. Due to ToBRFV’s ability to overcome the known tobamovirus resistance in tomato plants, studies looking for new genetic resistance to the virus were initiated. Indeed, recently, several ToBRFV-tolerant tomato genotypes and a single resistant genotype were identified [25].

To measure the effect of ToBRFV on the yield of tomato plants, the yield performance of ToBRFV-inoculated plants was compared with that of noninoculated plants of the same genotype. Two near-isogenic tomato lines were compared, Moneymaker and Moneymaker harboring the ToMV-resistance gene *Tm-2*^2^. Depending on the climate and cultivation practices, ToBRFV infection induced yield reductions of 19–55% regardless of the presence of the *Tm-2*^2^ resistance gene [27]. Due to the threat this emerging virus poses to global tomato cultivation, the USA issued a federal order regarding importing and inspecting tomato seeds, and the European Union has declared a quarantine status for ToBRFV [28,29].

As ToBRFV was identified only a few years ago, little is known regarding its interactions with the tomato host. Hence, we designed experiments to follow ToBRFV’s presence in the reproductive tissues of tomato plants using fluorescent in situ hybridization (FISH), confocal microscopy, and RT-PCR.

## 2. Materials and Methods

### 2.1. Virus

A field isolate of ToBRFV (GeneBank Acc. No. KXG619418) [3] was propagated in Moneymaker tomato plants that carried the *Tm-2*^2^ resistance gene (MM + *Tm-2*^2^) and were maintained in an insect-proof greenhouse. The culture was renewed every 3 to 4 weeks by mechanical inoculation; leaves of ToBRFV-infected tomato source plants were ground in 0.01 M phosphate buffer (pH 7.0) and applied to carborundum-dusted test plants. The carborundum was washed out, and the test plants were kept in a greenhouse.

### 2.2. Inoculation Using Infected Seeds or Pollen as Inoculum

Infected seeds—50 infected tomato seeds were ground in 2 mL 0.01 M phosphate buffer (pH 7.0) and applied to carborundum-dusted tobacco (*Nicotiana tabacum cv.* Xanthi-NN) test plants. The plants were followed for the appearance of virus-induced local lesions. ToBRFV’s presence in the local lesions was confirmed by RT-PCR.

Infected pollen—pollen powder (9 mg) was ground in 0.01 M phosphate buffer (pH 7.0) and applied to carborundum-dusted tomato test plants. Inoculated plants were maintained in an insect-proof greenhouse and monitored for symptoms. Following symptom appearance, ToBRFV’s presence was confirmed with RT-PCR.

### 2.3. Plant Material

Forty tomato plants (MM + *Tm-2*^2^) were grown in a greenhouse, inoculated with ToBRFV at the age of five leaves, and grown for fruit. Following inoculation, the plants were monitored for the appearance of disease symptoms and tested with RT-PCR for the presence of ToBRFV. These plants served as the source for ToBRFV-infected tissue, flower, pollen, fruit, and seeds. Noninoculated plants of the same genotype were grown in a separate greenhouse and served as control.

### 2.4. Viral Detection by RT-PCR

Total RNA from fresh leaves was extracted using a Viral RNA Extraction Kit (Bioneer, Daejeon, Korea) according to the manufacturer’s instructions.

Total RNA from infected and healthy tomato seeds was extracted after soaking the seeds in PBS buffer for two h. The seeds were then frozen with liquid nitrogen and ground using a mortar and pestle. Total RNA was extracted using the QIAMP Viral RNA Mini Kit (Qiagen, Hilden, Germany) according to the manufacturer’s instructions.

Total RNA extraction from infected and healthy tomato pollen grains was performed following Levin and Gilboa [30]. The nucleic acid concentrations from the extractions were measured using a spectrophotometer (NanoDrop ND-1000, Wilmington, NC, USA).

RT-PCR analysis was performed to detect ToBRFV accumulation in plants, seeds, and pollen. The first-strand cDNA was synthesized using complementary primers specific to ToBRFV with Revertaid^TM^ First-Strand cDNA Synthesis Kit (Thermo Scientific, Wilmington, NC, USA) according to the manufacturer’s instructions, and the resulting cDNA was used as a template for standard PCR. PCR conditions were 94 °C for 3 min, followed by 30 cycles of 94 °C for 30 s, 57 °C for 30 s, and 72 °C for 1 min followed by 10 min at 72 °C. Two sets of primers were used for ToBRFV detection: TBR-F-5738 (5′-GCAATTTGTGTTTTTGTCATC-3′) and TBR-R-6190 (5′-TTTAAGCATCTCGATTATCTCA-3′), amplifying a 474-nt fragment, or TBR-F-5556 (5′-GTTTAGTAGTAAAAGTGAGAATAATAG-3′) and TBR-R-6232 (5′-GTTTGCAGACACAATCTGTTATTTAAG-3′) amplifying a 676-nt fragment.

### 2.5. Fluorescent In Situ Hybridization (FISH)

Tomato flowers were sampled from 15 infected and 15 noninfected plants upon maturation. In each sampling round, we analyzed at least 10 flowers from each plant. The plants were sampled 10 different times; hence, approximately 1500 flowers from ToBEFV-infected plants were analyzed.

FISH was performed following Shargil et al. [31] using a ToBRFV-specific single-stranded DNA primer labeled with the fluorophore cyanine-Cy3 at the 3′ end (TBR-Cy3-6190; 5′-TTTAAGCATCTCGATTATCTCA/3Cy3Sp/-3′) corresponding to ToBRFV nucleotides 6190-6212.

Tissue (leaf and flower) samples from ToBRFV-infected and control non-infected plants were hand-sectioned and then fixed overnight at 25 °C in Carnoy’s fixation buffer (6:3:1 *v/v* mixture of chloroform: ethanol: glacial acetic acid). Samples were decolorized twice for 1 h at 25 °C in bleaching solution (6% H_2_O_2_ in ethanol) followed by pre-hybridization for 1 h at 25 °C in hybridization buffer (20 mM Tris-HCL, pH 8.0, 0.9 M NaCl, 0.01% sodium dodecyl sulfate, 30% formamide). The samples were hybridized overnight at 25 °C in hybridization buffer containing the fluorescent probe (100 pmol/mL). Samples were examined using a confocal microscope (Olympus 1X81/FV500, Tokyo, Japan).

### 2.6. 4′,6-Diamidino-2-phenylindole Dihydrochloride (DAPI) Staining

Following the FISH reaction (see above), samples were stained with 10 µL DAPI (0.1 µg/mL) solution (Thermo Scientific, Wilmington, NC, USA) according to the manufacturer’s instructions and examined using a confocal microscope.

### 2.7. Plant Pollination

A total of 21 clean mother plants (MM + *Tm-2*^2^) were grown from seeds in isolation in 15-L buckets in a greenhouse. To verify no ToBRFV infection, the plants were tested with RT-PCR. Once the plants started to flower, flowers were manually emasculated and hand pollinated two days later with pollen collected from ToBRFV-infected plants grown in a separate greenhouse. Eleven rounds of hand pollination were performed during a period of five weeks. Fruits were picked following maturation, and seeds were extracted from each fruit and stored separately at room temperature. At the end of the experiment, each mother plant was tested again for ToBRFV by RT-PCR.

### 2.8. Germination of Pollen Grains

Thirty tomato plants (MM + *Tm-2*^2^) were used for the experiment. Of those, 15 30-day-old tomato plants were inoculated mechanically with ToBRFV and grown for 90 days in a greenhouse. Another 15 non-inoculated plants were mock inoculated and grown as control. Ten flowers were collected from each plant on three dates: 45, 60, and 65 days after inoculation.

For the pollen germination testing, the anthers were removed from the collected flowers and soaked into a 1.5 mL germination solution (3.2 mM H_3_BO_3_, 5.8 mM CaNO_3_, 3.3 mM MgSO_4_, 1.9 mM KNO_3_, and 290 mM crystallized sucrose). The anthers were vortexed for 1 min to separate the pollen grains from the anther. Then, 100 µL of the pollen mixture was placed on a glass microscope slid covered with a thin layer of solid germination medium (2% *w*/*v* agarose dissolved in germination solution and allowed to solidify) and allowed to germinate for 90 min at room temperature. Pollen grains were visualized in a light microscope (Nikon, ECLIPSE NI-E), the number of germinated pollen grains was analyzed by IMEGEJ software.

### 2.9. Seed Harvesting

Seeds were harvested from ToBRFV-infected MM + *Tm-2*^2^ plants. Two-month-old plants were mechanically inoculated with the virus and grown to fruit maturity in the greenhouse. Seeds were harvested from the ripened tomato fruits; specifically, the fruit was cut in half, and all the seeds were removed, transferred to a small container, and left to ferment overnight at room temperature. The seeds were then transferred to a sieve, washed with water, and allowed to dry at room temperature. The seeds of each fruit were stored at room temperature in separate bags until use.

### 2.10. Statistical Analyses

Statistical analyses were carried out using Student’s *t*-test (SAS Institute Inc., Cary, NC, USA).

## 3. Results

### 3.1. ToBRFV Can Penetrate Tomato’s Reproductive Tissues

Tomato test plants (cv. Moneymaker harboring the *Tm-2*^2^ gene) were grown from seed and inoculated with ToBRFV at the age of five leaves. The plants developed typical disease symptoms that included pronounced mosaic on the leaves and leaf elongation at 20–25 days post-inoculation (DPI). ToBRFV’s presence in the inoculated plants and absence in the control plants was validated with RT-PCR.

Using the FISH technique, we followed ToBRFV’s presence in the vegetative and reproductive tissues. Figure 1 shows a close-up view of a tomato flower, containing sepals and petals as well as female and male organs (Figure 1). ToBRFV’s presence was first identified in the vegetative organs. No signal was seen in the healthy uninfected leaf (Figure 2A), while a strong fluorescent signal was observed in the leaf epidermis, parenchyma, and trichomes of the infected leaf (Figure 2B,C). In the infected plants, the virus was found in all reproductive organs. Compared with control virus-free flowers, the infected sepals, petals, ovaries, stamens, style, and stigma showed intensive fluorescent signals (Figure 2C–F). In some instances, at lower magnification, a fluorescent signal was visible in the plant’s ovule. However, observations at higher magnification clearly demonstrated that the virus was present in the ovary walls, pericarp, and placental tissues surrounding the ovules but did not penetrate the ovules (Figure 2F)

The virus was not found in the male reproductive organs of noninfected plants (Figure 3A), while a strong fluorescence signal was detected in the anthers and the pollen grains of the infected ones (Figure 3). However, close-up observations of the pollen grains indicate that only a certain number of the grains are infected (Figure 3B,C). Strong virus presence was also observed in the stigmas of the infected plants (Figure 3D,E).

To confirm that the virus can penetrate the pollen grains, we used double staining: first, we stained the pollen grains with DAPI, which stains the pollen nuclei in blue, followed by FISH to label the virus. The pollen of control plants did not produce any FISH signal, while in the infected plants, the co-localization of both the blue signal (DAPI stained nuclei) and the red signal (labeled viral RNA) confirmed virus penetration to the pollen grains and cytoplasmic localization (Figure 3J). RT-PCR analysis confirmed the presence of ToBRFV in pollen harvested from infected plants (not shown).

In addition, the virus-contaminated pollen was also tested in a bioassay; the pollen was crushed and used to mechanically inoculate eleven tomato test plants. All the test plants became infected within 21 days and showed typical ToBRFV symptoms. The presence of ToBRFV in the inoculated test plants was also validated with RT-PCR.

RT-PCR analysis of the fruits and seeds harvested from the infected plants confirmed their contamination with the virus. The seed-contaminated virus was infective, as was shown by a seed bioassay; seeds were crushed and used to inoculate tobacco plants, and the inoculated plants showed evident ToBRFV-induced local lesions and were positive on RT-PCR (Figure 4).

### 3.2. Cross-Pollination of Virus-Free Mother Plants with ToBRFV-Infected Pollen

Since the virus can penetrate at least some of the pollen grains and was found to be infective in a bioassay, we investigated the possibility of virus transmission during the pollination and fertilization processes. To this end, twenty-one virus-free mother tomato test plants were hand pollinated with pollen harvested from ToBRFV-infected plants and grown to fruit. The pollinated mother plants and the harvested fruits and seeds were tested for the presence of ToBRFV using RT-PCR (Table 1). The experiment was repeated twice. In the first experiment, 7 mother plants were found to be infected with the virus, while 14 were not (Table 1). While fruits and seeds harvested from the infected plants were heavily contaminated with ToBRFV, none of the fruits and seeds harvested from the noninfected mother plants contained ToBRFV. We were concerned that the seven infected mother plants were accidentally inoculated mechanically while the plants were handled during the pollination cycles. Thus, we performed the experiment a second time, taking care to prevent any accidental mechanical inoculation of the plants while handling the plants during pollination. As a result, all the mother plants were found to be virus-free at the end of the experiment when tested for the presence of ToBRFV by RT-PCR. Concomitantly, all the fruits and seeds harvested from these mother plants were also devoid of ToBRFV, as tested with RT-PCR (Table 1).

### 3.3. ToBRFV’s Effect on Pollen Germination

To investigate the effect ToBRFV infection might have on pollen germination, pollen grains collected from fifteen infected and fifteen noninfected tomato plants were germinated under controlled conditions. Approximately 12,000 pollen grains per treatment were allowed to germinate for 90 min in germination medium, and the number of germinated grains was counted (Table 2). The average germination rate of pollen grains collected from the noninfected plants was 73%, while the germination rate of pollen grains collected from infected plants dropped to 48.8% (Table 2). Therefore, the virus instigated a 33% reduction in pollen germination ability.

Figure 3 also shows that only a portion of the pollen grains was infected with the virus. Thus, we examined using FISH the percent infectivity in ca. 2800 pollen grains collected from ToBRFV-infected plants. It was found that the infection rate of pollen grains by ToBRFV was not high, with an average of 3.1% infectivity (Table 3).

Next, we assayed the germination rate of ToBRFV-infected pollen grains. Pollen grains collected from infected plants (the same pollen grains described in Table 3) were allowed to germinate in medium under controlled conditions. Double staining of the pollen grains confirmed normal germination of the noninfected pollen, while the infected grains were unable to germinate (Figure 5).

## 4. Discussion

ToBRFV is an emerging virus that threatens tomato production worldwide. The virus is seed-borne and was found in the tomato fruit [18,19,20,21,22], but little is known about the virus’s infectious routes within the tomato plant. To understand the infection processes of tomato reproductive tissues, we followed the viral genomic RNA using the FISH technique. Our findings show that the virus can penetrate nearly all reproductive organs of the tomato flower: petals, ovary, stamen, style, stigma, anther, and pollen grains. The virus was detected in the tissue surrounding the ovules in the ovary but was not detected inside the ovules. This is in agreement with previous studies demonstrating that the virus contaminating tomato seeds is localized in the seed coat and in the endosperm but is not found in the embryo [18,19,22].

Our results show that the virus significantly affects tomato pollen. Although the amount and morphology of pollen produced by the infected plants were not affected, pollen germination capability was affected. Only 49% of the pollen collected from ToBRFV-infected plants was able to germinate, compared with 73% of the pollen collected from the uninfected plants: the virus induced a 33% reduction in pollen’s ability to germinate.

It is well known that tomato, as well as other plants, is sensitive to biotic stresses, especially to heat stress. Night temperatures over 26 °C are known to reduce tomato yield, mainly due to the sensitivity of the developing pollen grains [32,33]. Our results demonstrate that a biotic stress such as a virus has a similar effect. ToBRFV can induce yield reductions of 19–55% in tomato [27], and this decrease can be attributed in part to the reduction in germination ability of pollen grains.

Contrary to the ovules, the virus was detected surrounding the pollen grains as well as inside the grains, with an average 3% infection rate of pollen grains. Thus, we tested whether the virus is pollen transmitted. Despite our finding that ToBRFV is able to penetrate some of the pollen grains, all the cross-pollinated noninfected mother plants that were virus-free by the end of the experiment gave rise to fruits and seeds that were devoid of ToBRFV (Table 1). It was also found that none of the infected pollen grains was able to germinate. Therefore, it was concluded that although ToBRFV can penetrate the pollen grains, it is not pollen transmitted.

TMV, the tobamovirus type member, was detected on the bodies of pollinating bumblebees and in pollen clumps taken from the bees’ bodies [34]. It was demonstrated that the pollinating bumblebees were able to transmit the virus from infected tomato to adjacent tomato plants in a greenhouse. It was suggested that as the bees hang from flowers by biting the anthers with their mandibles, they can mechanically transmit the virus that is attached to their bodies or carried with pollen clumps. It was also suggested that bees can transmit by contact ToMV and other sap-transmitted viruses, as indeed was demonstrated later with the greenhouse transmission of CGMMV from infected cucumber and melon plants to noninfected plants by pollinating honeybees (*Apis mellifera*) [35]. It was suggested that the virus is absorbed physically to the honeybees and is transmitted mechanically while the honeybees are foraging [35]. Recently, ToBRFV transmission by bumblebee in greenhouse-grown tomato plants was demonstrated [23]. Again, it was suggested that like TMV, ToBRFV transmission is by contact, either by transfer of infected-crude sap or mechanically during buzz pollination [23]. It should be noted that the accidental transmission of TMV, ToMV and a few other viruses by beetles, grass hoopers, insects and even birds has been documented [15,34,36].

It has been argued that CGMMV, a cucurbit-infecting tobamovirus, is pollen transmitted [35]. The virus is known to be seed-transmitted in several cucurbit crops such as cucumber, melon, watermelon, and squash [16]. Artificial pollination experiments were performed in which male flowers were collected from CGMMV-infected cucumber plants and rubbed against the stigmata of the female flowers of uninfected cucumber plants. Fruits collected from these plants became infected at a range of 17–50%, and seeds harvested from these fruits were infected with CGMMV [37]. However, the authors could not exclude the possible transmission by mechanical inoculation due to contact while rubbing the infected male flowers [37]. In a later study, using FISH to follow CGMMV RNA in infected cucumber and melon plants, anther tissue of the male flowers was found to be heavily infected, but pollen grains were found to be virus-free [31].

Our results do not indicate how the virus penetrates the pollen grains. One possibility is that the virus is only able to penetrate defective pollen grain that has already lost its ability to germinate regardless of the virus. Alternatively, as the virus induces morphological changes such as leaf deformation in infected plants, perhaps at the stage of flower development, the virus can cause structural changes to the developing pollen grains. Such formations will allow the virus to penetrate the pollen cells in an unknown mechanism, and as a result of the virus activity, the pollen grain loses its ability to germinate. Nonetheless, to the best of our knowledge, this is the first demonstration of a tobamovirus that is able to penetrate tomato pollen grains and induces the germination arrest of pollen.

## Figures and Tables

**Figure 1 cells-11-02864-f001:**
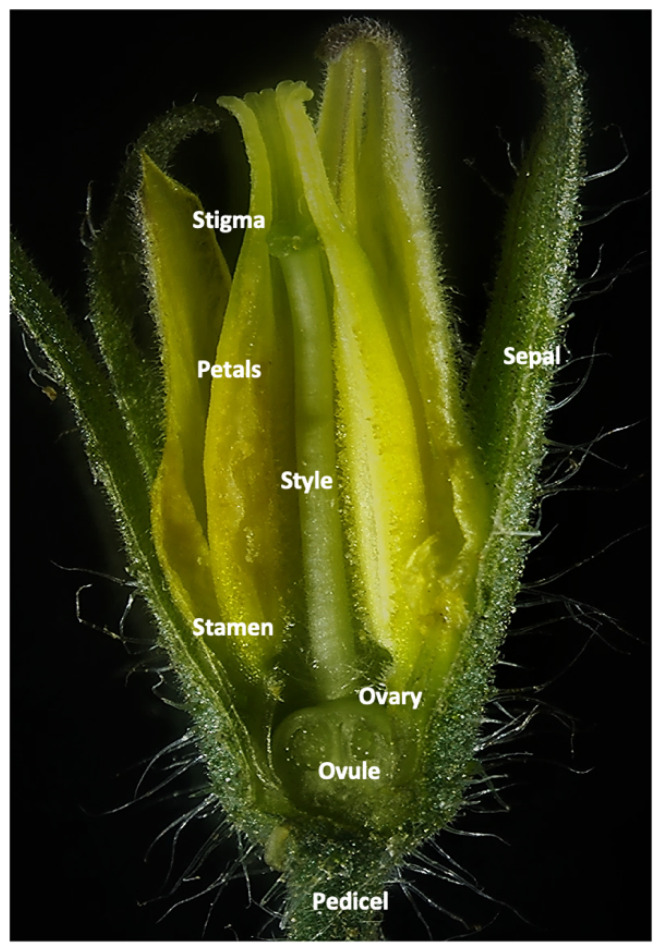
Structure and organs of a tomato flower.

**Figure 2 cells-11-02864-f002:**
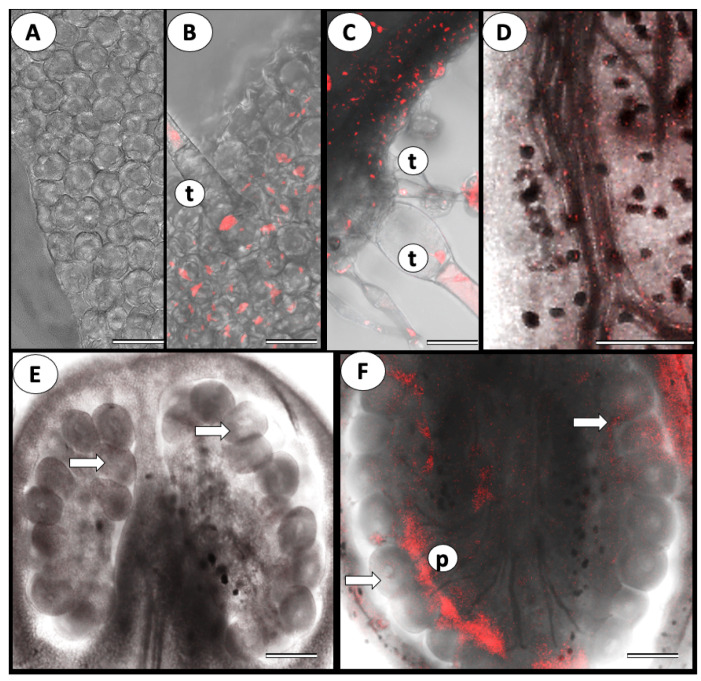
Representative confocal microphotographs of ToBRFV presence in the vegetative and reproductive organs of tomato. In situ hybridization with a specific ToBRFV DNA probe fluorescently labeled with Cy3. Intact uninfected plants served as control. (**A**) Leaf surface in control plant, Bar = 50 µm; (**B**) Leaf surface in infected plant. Note numerous red dots (fluorescent signal) in the epidermal cell and trichomes (t), Bar = 50 µm; (**C**) Transverse section of the infected leaf. The signal is visible in all leaf tissues and trichomes (t), Bar = 100 µm; (**D**) Section of the sepals of infected plant; note fluorescent signal (small red dots), Bar = 200 µm; (**E**) Cross-section of the control uninfected ovary; ovules (arrows) are visible, Bar = 200 µm. (**F**) Cross-section of the infected ovary; ovules (arrows) are not infected, but ovary walls, pericarp, and placental tissues (p) show fluorescence signal of virus infection. Bar = 200 µm.

**Figure 3 cells-11-02864-f003:**
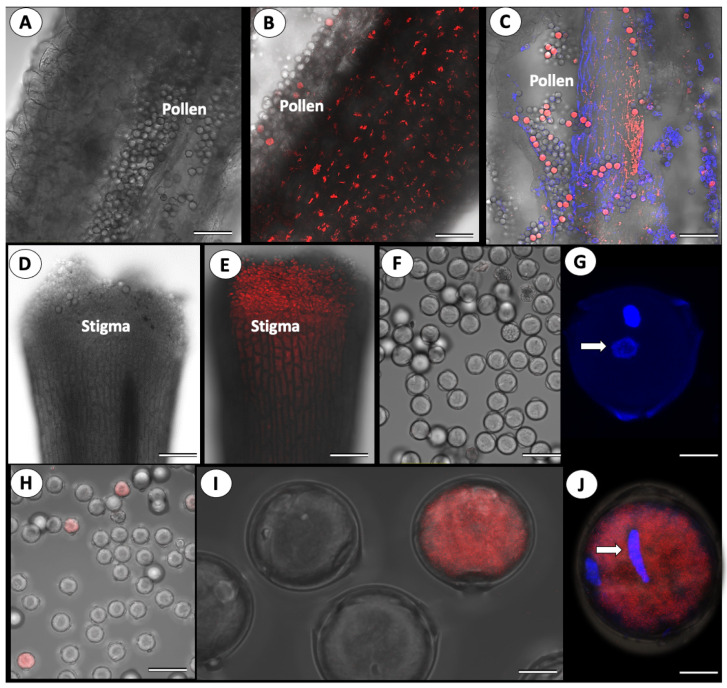
Representative confocal microphotographs of ToBRFV presence in the flower organs of tomato. In situ hybridization with a specific ToBRFV DNA probe fluorescently labeled with Cy3. Intact uninfected plants served as control. (**A**) Transverse section of the anther of control plant. Clean pollen and anther tissues are visible, Bar = 100 µm; (**B**) Transverse section of the anther of the infected plant. Note fluorescent signal in the anther tissue, pollen is partly infected, Bar = 100 µm; (**C**) Transverse section and double staining with DAPI and Cy3 of the infected anther. Fluorescence signal (red dots) is visible in several pollen grains, Bar = 200 µm; (**D**) Transverse section of control style, Bar = 200 µm; (**E**) Transverse section of infected style. The stigma of the infected flower looks heavily infected. Bar = 200 µm; (**F**) Pollen grains of control plant, Bar = 200 µm; (**G**) Close-up of the pollen grain of the control plant, double stained with DAPI and Cy3. Note the absence of red fluorescence signal (viral infection). The pollen nucleus is stained blue (arrow). Bar = 10 µm; (**H**) Close-up of the pollen grains in the infected plants. Bar = 20 µm; (**I**) Pollen grains of infected plants, Bar = 50 µm; (**J**) Close-up of the pollen grain of the infected plant, double staining with DAPI and Cy3. Note red fluorescence signal (viral infection). The pollen nucleus is stained blue (arrow). Bar = 10 µm.

**Figure 4 cells-11-02864-f004:**
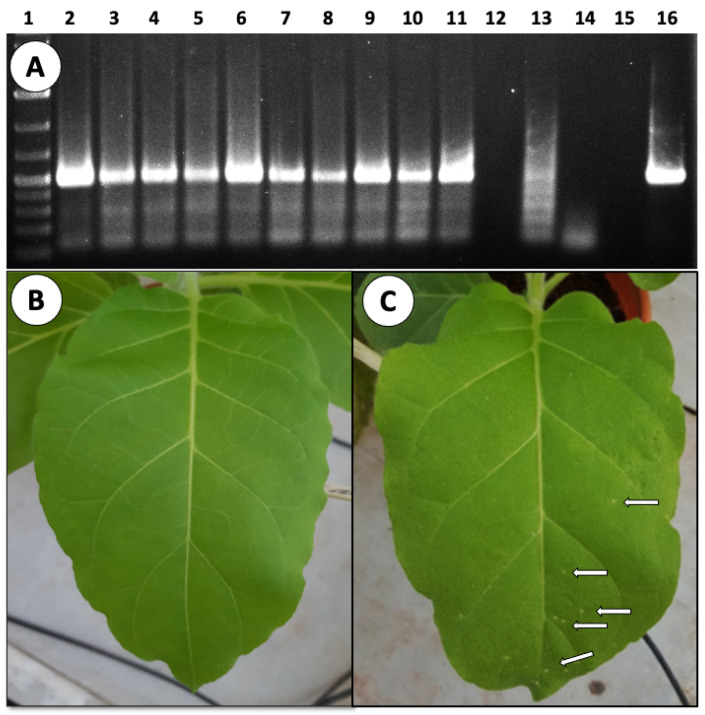
PCR analysis of tomato seeds for ToBRFV presence and infectivity test for seed-associated ToBRFV. (**A**) RT-PCR detection of ToBRFV in seeds harvested from infected plants. Each lane represents 15 seeds harvested from a different plant. Lanes: (1) Molecular weight markers; (2–11) Seeds harvested from infected plants; (12, 15) empty lanes; (13) seeds harvested from a noninoculated plant; (14) reaction mix; (16) Total RNA from an infected leaf. (**B**,**C**) Tobacco leaf (*N. tabacum* cv. Xanthi–NN) was inoculated with a crude extract from 50 seeds harvested from healthy tomato (**B**) or from an infected tomato plant (**C**). Note the presence of local lesions (arrows). ToBRFV’s presence in the developed local lesions was validated by RT-PCR. Pictures were taken 72-hr after inoculation.

**Figure 5 cells-11-02864-f005:**
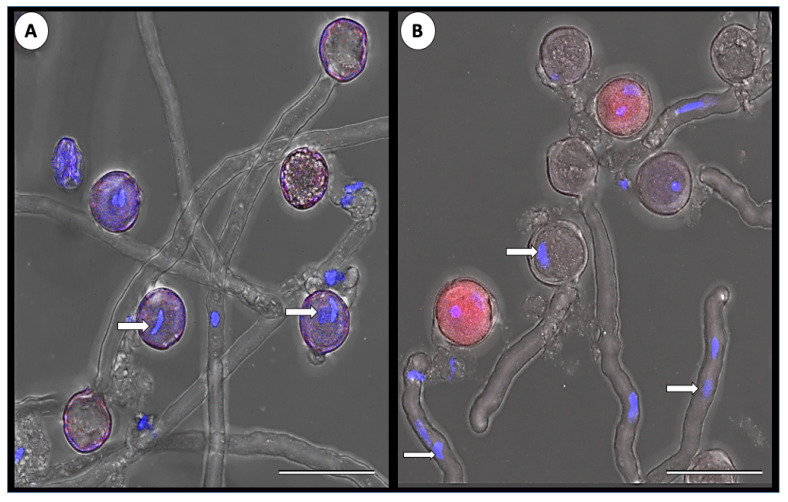
Representative confocal microphotographs of ToBRFV presence in germinating pollen grains of tomato. Pollen grains were allowed to germinate for 90 min, followed by DAPI staining and in situ hybridization with a specific ToBRFV DNA probe fluorescently labeled with Cy3. (**A**) Germination of pollen from control noninfected plant; nuclei stained in blue (DAPI) are visible in the pollen grain and tubes (arrows), Bar = 50 µm; (**B**) Germination of pollen from infected plant. Note that the infected grains do not germinate (stained in red). In germinating noninfected grains, nuclei are stained in blue (DAPI) and are visible in the grain and pollen tubes (arrows), Bar = 50 µm.

**Table 1 cells-11-02864-t001:** ToBRFV transmission by the cross-pollination of noninfected tomato mother plants with pollen from infected plant.

Ex.	No. of Mother Plants		No. of Fruits		No. of Seeds *		Infection (%)	
	Infected **	Not infected	Infected	Not infected	Infected	Not infected	Fruit	Seeds
I	7		20	1	39	3	95.2	92.8
		14	0	42	0	96	0.0	0.0
II	0							
		21	0	62	0	930	0.0	0.0

* In Experiment I, No. of seeds is the total number of seeds produced, as not all fruit gave seeds. In Experiment II, 15 seeds per fruit were tested. ** At the end of the experiment, all the mother plants were tested for the presence or absence of ToBRFV with RT-PCR.

**Table 2 cells-11-02864-t002:** ToBRFV’s effect on pollen germination competence.

	Infected Plants				Non-Infected Plants			
	No. of Pollen Grains				No. of Pollen Grains			
Plant No.	Germinated	Not Germinated	Total	Germination (%)	Germinated	Not Germinated	Total	Germination (%)
1	299	279	578	51.7	646	247	893	72.3
2	816	533	1349	60.5	931	461	1392	66.9
3	243	270	513	47.4	567	139	706	80.3
4	383	358	741	51.7	648	230	878	73.8
5	359	381	740	48.5	534	240	774	69.0
6	470	329	799	58.8	316	167	483	65.4
7	693	470	1163	59.6	548	335	883	62.1
8	899	972	1871	48.0	880	233	1113	79.1
9	158	232	390	40.5	612	95	707	86.6
10	174	183	357	48.7	584	176	760	76.8
11	581	557	1138	51.1	254	131	385	66.0
12	618	679	1297	47.6	1021	306	1327	76.9
13	192	233	425	45.2	338	129	467	72.4
14	107	150	257	41.6	579	164	743	77.9
15	212	470	682	31.1	504	193	697	72.3
Total	6204	6096	12,300		8962	3246	12,208	
				Av: 48.8 ± 2.0 *				Av: 73.2 ± 1.7 *

* Significant at *p* < 0.0001. ±Standard Error.

**Table 3 cells-11-02864-t003:** Infection of pollen grains harvested from ToBRFV-infected plants.

		No. of Pollen Grains		
Experiment	Field/Slide No.	Total	Infected	Infectivity (%)
I	1	32	1	3.1%
	2	31	2	6.5%
	3	240	6	2.5%
	4	238	8	3.4%
	5	188	6	3.2%
	6	28	1	3.6%
	7	51	3	5.9%
	8	93	5	5.4%
	9	73	1	1.4%
	10	78	1	1.3%
	11	76	1	1.3%
II	1	103	3	2.9%
	2	54	5	9.3%
	3	56	4	7.1%
	4	223	2	0.9%
	5	16	1	6.3%
III	1	43	1	2.3%
	2	151	2	1.3%
	3	74	1	1.4%
	4	49	1	2.0%
	5	28	1	3.6%
	6	106	1	0.9%
	7	109	1	0.9%
	8	117	2	1.7%
IV	1	403	1	0.2%
	2	171	2	1.2%
Total:	26	2865	68	
				Av: 3.1%

## Data Availability

Data in this study are available from the authors upon request.

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
