# Peer review of "ToBRFV Infects the Reproductive Tissues of Tomato Plants but Is Not Transmitted to the Progenies by Pollination"

_cells, 2022, doi:10.3390/cells11182864_

Round 1
Reviewer 1 Report
In this manuscript, the authors investigated the distribution of ToBRFV in the reproductive tissues of tomato by the fluorescent in situ hybridization (FISH) and RT-PCR techniques. They found that the virus was detected in the leaves, petals, ovary, stamen, style, stigma, and pollen grains, but not inside the ovules. ToBRFV could affect the pollen germination capability but could not be not transmitted to the progenies by pollination.
There are several issues needed to be addressed:
1.Section 3.1 lack of the typical disease symptoms of ToBRFV.
2.Figure 4, the inoculated tobacco leaves should be confirmed by RT-PCR.
3.How many ovary was detected in Figure 2? Whether ToBRFV exist in the embryo of the mature seed.
4.Did the authors test other varieties of tomato, such as variety lacking tobamovirus resistance Tm-2. Maybe the Tm-2 gene have certain influence on pollen transmission
Author Response
Response to Reviewer’s comments
Authors: we would like to thank the reviewers for constructive comments. Please find our response below under each specific comments.
Reviewer 1
In this manuscript, the authors investigated the distribution of ToBRFV in the reproductive tissues of tomato by the fluorescent in situ hybridization (FISH) and RT-PCR techniques. They found that the virus was detected in the leaves, petals, ovary, stamen, style, stigma, and pollen grains, but not inside the ovules. ToBRFV could affect the pollen germination capability but could not be transmitted to the progenies by pollination.
There are several issues needed to be addressed:
1.Section 3.1 lack of the typical disease symptoms of ToBRFV.
Authors: A short sentence describing ToBRFV symptoms was added to section 3.1, line 218.
2.Figure 4, the inoculated tobacco leaves should be confirmed by RT-PCR.
Authors: Presence of ToBRFV in the inoculated tobacco leaves was indeed validated by RT-PCR. It was stated in Materials and Methods (line 139), and in the results (line 219). Apparently it wasn’t clear enough. A sentence stating that was added to section 2.2, line 117, and to the legend of figure 4, line 290.
3.How many ovary was detected in Figure 2? Whether ToBRFV exist in the embryo of the mature seed.
Authors:. Figure 2 shows two ovaries, one from uninfected and another from infected plant. In each sampling we have analyzed at least 10 flowers from each of 30 plants – 15 infected and 15 control uninfected. That is 300 flowers per sampling. The plants were sampled 10 times during the experiment. Hence, we have analyzed at least 1500 flowers from ToBRFV-infected plants during this study. Indeed, these data were missing and were added to section 2.5 lines 156-159.
We did not analyze the presence or absence of ToBRFV in the embryo of mature seeds. However, it was tested in other studies and the virus was not found in the embryo (Klap et al., 2020; Salem et al 2021; Davino et al., 2020). This point is discussed in the introduction (lines 62-63).
4.Did the authors test other varieties of tomato, such as variety lacking tobamovirus resistance Tm-2. Maybe the Tm-2 gene have certain influence on pollen transmission.
Authors: In this study we tested only a tomato variety harboring Tm-2. As most if not all commercial tomato hybrids today carry the Tm-2 gene for ToMV-resistance, and as ToBRFV overcomes this resistance, we tested the effect ToBRFV has on a model tomato genotype with Tm-2. Moreover, results show that Tm-2 targets the MP of ToMV, and that ToBRFV MP doesn’t interact with it (Hak and Spiegelman, 2021). We doubt very much that Tm-2 effects tomato pollen germination, as it has been used by seed companies for over 50 years. Nonetheless, we agree that the effect of ToBRFV on pollen from a tomato genotype without Tm-2 should be tested in a further study.
Reviewer 2 Report
figure 4 why you choose tobacco as material , I think tomato is better.
some reported ToBRFV can transmitted via seed. please discuss in disscussion. method is not detail.
Author Response
Authors: we would like to thank the reviewers for constructive comments. Please find our response below under each specific comments.
Reviewer 2
Figure 4 why you choose tobacco as material, I think tomato is better.
Authors: we have tried to use tomato as test plants. It turns out that the tomato plants are too delicate for mechanical inoculation from seed extracts. The damage to the tomato leaves was such that the plants nearly died following inoculation. We have consulted colleagues from a seed company who informed us that it doesn’t work with tomato and therefore we should use tobacco plants.
Some reported ToBRFV can transmitted via seed. Please discuss in disscussion.
Authors: The reports regarding ToBRFV via seeds were discussed at length in the introduction, lines 61-74. A sentence to that effect was added to the discussion, lines 355-357.
Method is not detail.
Authors: We have added details to sections 2.2 and 2.5.
Reviewer 3 Report
The research article on “ToBRFV infects the reproductive tissues of tomato plants but is not transmitted to the progenies by pollination " focuses mainly on the transmission study of Tomato brown rugose fruit virus in tomato host plants. Authors proved that ToBRFV might infect reproductive organs and pollen grains of tomatoes, but is not pollen transmitted. The investigation/ current study can be much interesting for the management of crop improvement. I have enjoyed reading the entire manuscript except for a few places with typo errors and some sentences are very long and not appropriate. This is timely credible work; however, authors should improve their data presentation and style wherever necessary. One minor concern that is required to be clear before acceptance:
I wonder if the authors checked the expression of the virus through Real-time PCR.
Author Response
The research article on “ToBRFV infects the reproductive tissues of tomato plants but is not transmitted to the progenies by pollination " focuses mainly on the transmission study of Tomato brown rugose fruit virus in tomato host plants. Authors proved that ToBRFV might infect reproductive organs and pollen grains of tomatoes, but is not pollen transmitted. The investigation/ current study can be much interesting for the management of crop improvement. I have enjoyed reading the entire manuscript except for a few places with typo errors and some sentences are very long and not appropriate. This is timely credible work; however, authors should improve their data presentation and style wherever necessary.
Authors: Thank you for the very positive review. We have tried to correct the presentation where we could, and hope it is satisfactory now.
One minor concern that is required to be clear before acceptance:
I wonder if the authors checked the expression of the virus through Real-time PCR.
Authors: Yes, it’s currently under investigation.
Round 2
Reviewer 1 Report
After reading the revised version, I think this paper is complete for publication.
Author Response
Thank you

Reviewer 2 Report
please explain the reason of why you choose tobacco leaf as material in Figure
This is in agreement with previous studies demonstrating that the virus contaminating tomato seeds is localized in the seed coat and in the endosperm, but was not found 344 in the embryo (18, 19, 22).
Reference 19, please confirm 19in right place.
Author Response
Please see attached response
